# Contextual Temperature for Language Modeling

## Abstract

This paper presents a generalized approach to provide an individual optimal temperature trajectory to each class. Temperature scaling is an effective technique to control the smoothness of the distribution. Current implementations of temperature scaling assume either a fixed or manually-crafted dynamically changing schedule. This schedule is used to identify the shared optimal trajectory, which is applied to every class. However, our studies indicate that under the change of context, we can identify the corresponding individual optimal trajectory for each class. To this end, we propose *contextual temperature*, a mechanism that allows the temperature to be learned during training along with the remaining model parameters, over the context for each vocabulary. Experimental results confirm that *contextual temperature* significantly improves state-of-the-art language models. The proposed method achieves a perplexity of 55.31 and 62.89 on the test set of Penn Treebank and WikiText-2, respectively. Additionally, in-depth analyses indicate that (a) every vocabulary possesses its individual schedule of temperature, and (b) due to the increased context from every time step, the optimal trajectory drops in order to suppress the uncertainties.

## 1 Introduction

Due to the discrete nature of human language, the Softmax layer is mandatory to convert learned representations into a sequence of linguistic vocabulary. Temperature scaling is often used along with the Softmax layer in natural language processing. Generally, the temperature is applied as a denominator to output logits of the Softmax layer (Krizhevsky et al., 2012; Bahdanau et al., 2014; Hu et al., 2017; Caccia et al., 2018). The scale of temperature controls the smoothness of the output distribution. That is, given a temperature $\tau$, when $\tau \to \infty$, the distribution becomes more uniform, thus increasing the uncertainty. Contrarily, when $\tau \to 0$, the distribution collapses to a point mass.

Although temperature scaling has been justified to achieve great success, existing implementations are limited. For instance, the temperature is assumed to be constant throughout training (Norouzi et al., 2016; Ma et al., 2017; Chen et al., 2019), or to be a fixed schedule (Hu et al., 2017). Most importantly, none of the existing works studies the effects on different word tokens when the temperature changes. In reality, however, the temperature of each token can be dramatically different. Figure 1(a) shows the optimized temperature for each word token during the course of training. As shown in the figure, some certain words have a distinct heating-up temperature scaling, while the majority of words have a scaling that gradually cooling down the temperature. We argue that existing methods limit themselves to some fixed schedules, and thus have great difficulty to generalize.

In addition, another example can be observed in Figure 1(b), which indicates that the average temperature drops, as the length of the context increases. This suggests that the temperature mechanism helps promote stochasticity in the beginning of a sentence, then gradually suppressing the uncertainty until the end. All of these suggest the use of a more generalized temperature mechanism with the advantage of being able to deal with these phenomena.

We propose *contextual temperature*, a generalized temperature scaling method which takes both the history of a sequence and a particular vocabulary into consideration. Through optimizing the use of temperature scaling by the change of contexts, contextual temperature has the ability to generate a unique temperature for each vocabulary. As we parameterize contextual temperature using a deep neural network, its parameters co-evolve with the rest of model parameters, making the temperature

adapt to the training procedure. Experiment results on language modeling demonstrate significant improvements on both Penn Treebank and WikiText-2 datasets, with and without the fine-tune setting. Consistent improvements are shown on both validation and testing splits. In addition, we conduct comprehensive analyses and ablation studies to confirm the improvements of contextual temperature. We observe that our method is capable of controlling the uncertainties as the patterns of contexts change, allowing language models to achieve much better performances.

To the best of our knowledge, this is the first systematic work that studies the role of per-token temperature changing over context for training language models. The experimental results suggest a new way of training sequential models with discrete outputs, that is, using parameterized temperatures. We have also established the link between the control of model uncertainty and the use of temperatures, paving the way for extensions on tasks that require such long-term control, for instance summarization, translation, and dialogue generation.

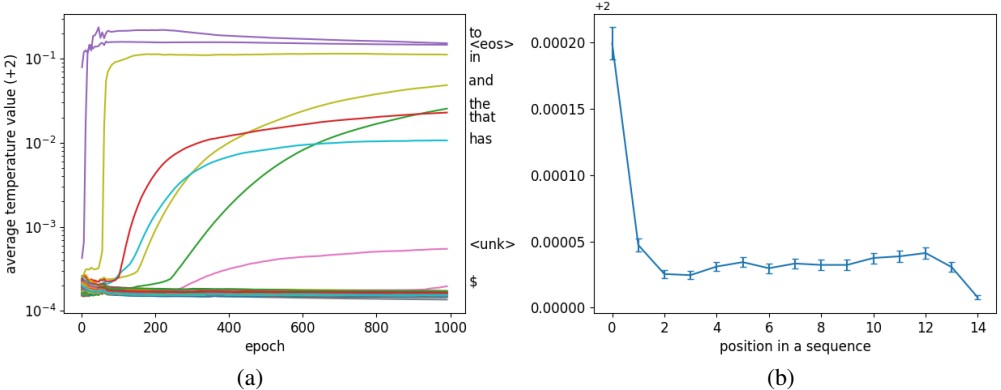

(a)                                         (b)

Figure 1: (a) Temperature of each word using the proposed method over training epochs. Each line in the figure represents a distinct token ranked by the frequency. The vertical axis shows the actual temperature value with a minus of 2. (b) Mean and confidence interval of the temperature over the position in the sentence. We can see that the average temperature is high at the beginning of the sentence and gradually decreases over time.

## 2 RELATED WORK

### 2.1 LANGUAGE MODELING

Given a sequence of history tokens $x_{1..t-1}$, language modeling aims at predicting the next token $x_t$ in a sentence. In other words, the goal is to model the probability of choosing the $k^{th}$ word, $P(x_t = k|x_{1..t-1})$, by using an encoding function $f$ and a Softmax layer $\sigma$. $f$ is parameterized by $\theta$ to compress the history $\mathbf{z} = f(x_{1..t-1}; \theta)^T \cdot W$.

$$P(x_t = k|x_{1..t-1}; \theta) = \sigma(\mathbf{z})_k \tag{1}$$

Here $W \in \mathbb{R}^{d \times K}$ is a matrix that converts a $d$ dimensional output of $f$ into a $K$ dimensional vector. The state-of-the-art language models adopt deep neural networks as the encoding function, including the earlier feed-forward networks (Bengio et al., 2003), more recent recurrent-based ones (Mikolov et al., 2010; Hochreiter & Schmidhuber, 1997) and attention-based models (Devlin et al., 2018).

### 2.2 SOFTMAX LAYER

A Softmax layer $\sigma(\mathbf{z})$ normalizes a $K$ dimension, real-valued vector $\mathbf{z}$ to make it sum to 1.

$$\sigma(\mathbf{z})_i = \frac{z_i}{\sum_j^K e^{z_j}} \tag{2}$$

Recent progress in language modeling suggests that Mixture of Softmaxes (MoS) (Yang et al., 2018) would significantly improve the performance by computing multiple Softmax distributions and use a set of weights to sum them up as the final probability distribution. To achieve this, a set of $M$ matrices $W_m$ is applied to the output generated by $f$. That is, $\mathbf{z}_m = f(x_{1..t-1}; \theta)^T \cdot W_m$, where $W_m \in \mathbb{R}^{d \times K}$ has the dimension of the embedding $d$ and the dimension of the vocabulary $K$. The probability distribution of the next word $x_t$ under the MoS model is thus a mixture of $M$ Softmaxes weighted by $\pi$. Here $\Theta = \cup_{m=1}^{M} W_m \cup \theta$.

$$P_{MoS}(x_t = k|x_{1..t-1}; \Theta) = \sum_{m}^{M} \pi_m \cdot \sigma(\mathbf{z}_m)_k \qquad (3)$$

## 2.3 Temperature Scaling

Temperature scaling is an approach used to control the smoothness of the distribution. Instead of applying the Softmax function as suggested in Equation 1, logits here are divided by the temperature $\tau$ before passing through the Softmax layer.

$$P(x_t = k|x_{1..t-1}; \theta, \tau) = \sigma(\mathbf{z}/\tau)_k \qquad (4)$$

**Constant Temperature.** Earlier works that adopt the constant temperature can be traced back to Model Distillation (Hinton et al., 2014). Several works have been taking advantage of a fixed temperature during training (Norouzi et al., 2016; Ma et al., 2017; Chen et al., 2019). For instance, (Norouzi et al., 2016; Ma et al., 2017) optimize log-likelihood on augmented outputs, which are sampled proportionally to their exponential scaled rewards, where the temperature controls the degree of augmentation. Other works incorporate the temperature to enhance calibration during inference (Guo et al., 2017), or to achieve a great trade-off between quality and diversity (Caccia et al., 2018).

**Dynamic Temperature Over Training Iterations.** A wide range of works adopt a manually-crafted schedule to tune the temperature during training. Notably in (Hu et al., 2017), a new text generation architecture is introduced that combines VAE and a discriminator. Since text samples are discrete and non-differentiable, a continuous approximation based on the Softmax with a decreasing temperature is used to enable gradient propagation from the discriminator to the generator. Similar techniques are adopted in the gumbel-softmax trick (Jang et al., 2017), which also allows gradients to pass through discrete sampling objectives. Studies also show that temperature with a heating up schedule makes the embedding vectors more compact (Zhang et al., 2018).

**Dynamic Temperature Over Word Position.** Another work that is closely related to our work, is the adaptive temperature over an attention model (Lin et al., 2018). We note that contextual temperature further learns the temperature for each vocabulary in the output distribution.

## 3 Methods

### 3.1 Contextual Temperature

Contextual temperature is a mechanism that chooses the optimal temperature by considering the "context" of a word $x_t$. A context of a word includes not only the history $x_{1..t-1}$, but also the specific vocabulary $k$ that we calculated the probability on. Such a mechanism allowed us to parameterize the temperature vector $\tau$ using a deep neural network and adapted the softness of the Softmax layer.

Our temperature vector $\tau \in \mathbb{R}^K$ was generated from the mapping function $f$ as discussed before. Although $f$ can be any sequential models such as RNN or LSTM, we chose to parameterize it by the AWD-LSTM (Merity et al., 2018). We omitted the details of its architecture due to the limited space in this paper. The output of AWD-LSTM was a vector with dimension $D$. We multiplied this vector by two matrices $W_{\tau_1} \in \mathbb{R}^{D \times Q}$ and $W_{\tau_2} \in \mathbb{R}^{Q \times K}$. Please note that one can potentially use a single matrix to represent these two. However, doing so can significantly increase the number of parameters and thus in practice we factorized them into two. Finally, we scaled temperatures using a Softmax function over the dimension of the vocabulary and its range was bounded in $[\frac{\alpha}{\beta}, \frac{1+\alpha}{\beta}]$.

$$\tau = \frac{\sigma(f(x_{1..t-1}; \theta)^T \cdot W_{\tau_1} \cdot W_{\tau_2}) + \alpha}{\beta} \qquad (5)$$

### 3.2 Contextual Temperature MoS

We then used the Contextual Temperature Mixture of Softmaxes architecture for language modeling (CT-MoS). The CT-MoS model extends Equation 3 by adding contextual temperature in Equation 5. Here $\oslash$ represents the element-wise division between $\mathbf{z}_m$ and the temperature vector $\tau$. The results after division are then sent to the Mixture of Softmaxes model. Since new parameters are added to the model, the CT-MoS now has a total of parameters $\Theta = \cup_{m=1}^{M} W_m \cup \theta \cup W_{\tau_1} \cup W_{\tau_2}$.

$$P_{CT-MoS}(x_t = k | x_{1..t-1}; \Theta) = \sum_m^M \pi_m \cdot \sigma(\mathbf{z}_m \oslash \tau)) \tag{6}$$

One thing worth noticing is that comparing to prior works, the proposed contextual temperature model has the ability to adopt a different temperature for (a) different vocabulary in the prediction, (b) different position of the same vocabulary given the history and (c) a tunable parameter that is capable of changing as the training progresses. The detailed architecture is illustrated in Figure 2, which highlights the difference between the proposed CT-MoS model and the MoS model.

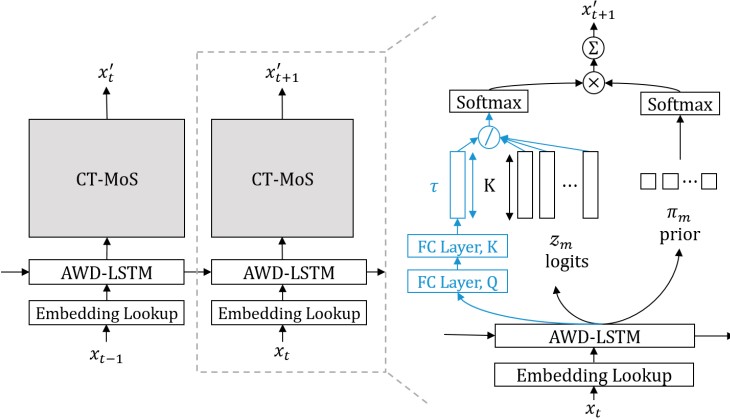

Figure 2: The architecture of the proposed CT-MoS model. Black components are those the same as the MoS model, while the blue ones are the newly added ones in our approach.

### 3.3 Training Objectives and Loss Scaling

We adopt the same regularization techniques in MoS (Yang et al., 2018) and AWD-LSTM (Merity et al., 2018). Our loss function thus consists of four terms: Cross Entropy ($\mathcal{H}$), Activation Regularization (AR), Temporal Activation Regularization (TAR), and Weight Decay (WD). AR is used to penalize high values of outputs, and TAR is used to prevent outputs from changing too much between timesteps. WD prevents the model from overfitting. Here $\gamma 1$, $\gamma 2$ and $\gamma 3$ are constants to scale regularization terms. And $m$ is the dropout mask (See in Merity et al. (2018)).

$$\mathcal{L}(\Theta) = \underbrace{\frac{\sum_i^K \tau_i}{K}}_{LS} \mathcal{H}(\hat{\mathbf{y}}, \mathbf{y}) + \gamma_1 \underbrace{L_2(m \odot f(x_{1..t-1}; \theta))}_{AR} + \gamma_2 \underbrace{L_2(f(x_{1..t-1}; \theta) - f(x_{1..t}; \theta))}_{TAR} + \gamma_3 \underbrace{L_2(\Theta)}_{WD} \tag{7}$$

The uniqueness in our setting is the Loss Scaling term (LS). The use of temperature scaling makes gradients of $\mathcal{H}$ disproportional to that of the other three terms, so the scale of $\mathcal{H}$ needs to be adjusted accordingly. A similar phenomenon is reported in (Hinton et al., 2014). In our case, only gradients of $\mathcal{H}$ will be influenced by temperatures. As for the other regularization terms, their gradients are not effected since they do regularization on parameters before temperature scaling. This difference leads to the unbalance between four objectives. Therefore, we scale $\mathcal{H}$ in order to redress the balance. We empirically find that scaling $\mathcal{H}$ with the average of temperatures works well in our setting.

## 4 EXPERIMENTS

### 4.1 DATASETS

We evaluate contextual temperature on two widely-used datasets for language modeling: Penn Tree-bank (PTB) (Marcus et al., 1993; Mikolov et al., 2011) and WikiText-2 (WT2) (Merity et al., 2017). The PTB dataset contains 929k training, 73k validation and 82k test tokens. The vocabulary size is capped at 10,000 most frequent unique tokens, while the rest of tokens are replaced with the <unk> token. We follow common practices to pre-process the dataset (Mikolov et al., 2011): (a) words are lower-cased, (b) numbers are replaced with "N", (c) newlines are replaced with <eos> and (d) punctuation is removed.

WikiText-2 is derived from Wikipedia articles and released as a popular option to train language models. WT2 contains 2M training tokens and a vocabulary of around 33k tokens. Compared to PTB, WT2 is roughly two times larger in sample size and three times larger in vocabulary size.

### 4.2 EXPERIMENTAL SETUPS

We conduct experiments on PTB and WT2 using one and four 1080 Ti GPUs, respectively. The environment we use is PyTorch (Paszke et al., 2017). We follow the training configurations as reported in the MoS paper and their github[1]. For both PTB and WT2, we use the same number of parameters as MoS. We use three layers of LSTM with embedding sizes of 960-960-620 in PTB experiments, that is, the number of embedding for functional mapping is $D = 620$. The embedding size here is $Q = 280$. For WT2, we use three layers of LSTM with embedding sizes of 1150-1150-650. In both experiments, $\gamma_1, \gamma_2, \gamma_3$ for the regularization terms are 2.0, 1.0 and $1.2e^{-6}$, respectively. The number of Softmaxes to be mixed is $M = 15$. Furthermore, we perform normalization as shown in Equation 5 to ensure that temperatures have positive values. We have tried several different values for $\alpha$ and $\beta$, and find $(\alpha, \beta) = (1, \frac{1}{2})$ works best in all experiments.

We use MoS as our baseline model, and AWD-LSTM as an additional baseline for comparison. Since MoS is the state-of-the-art model on PTB and WT2 datasets for language modeling, attention based models, such as Transformer-XL (Dai et al., 2019) and GPT-2 (Radford et al., 2019), are not used here to be the baseline. However, the performances of CT-MoS, MoS, Transformer-XL and GPT-2 is listed in Appendix A.

### 4.3 RESULTS

First, we show experimental results on the PTB dataset in Table 1. The original MoS model (Yang et al., 2018) has a model size of 22M parameters. To make a fair comparison, we augment the number of parameters of the MoS model to have 24M parameters and name it Mos$^+$. This is done by increasing the size of word embedding from 280 to 410. We show that our CT-MoS model outperforms AWD-LSTM, MoS and MoS$^+$ models on both validation and test sets with and without fine-tuning (Merity et al., 2018) and dynamic evaluation (Krause et al., 2017). Our best model achieves $48.12$ perplexity on the validation set and $47.42$ on the test set, beating the state-of-the-art MoS model with a significant margin. Table 2 provides results for WT2, a much larger language modeling dataset. We see a similar pattern to the PTB dataset that CT-MoS outperforms the state-of-the-art models. When using dynamic evaluation, our model also achieves comparable results to MoS and AWD-LSTM.

### 4.4 ABLATION STUDIES

**Fixed Model Size Comparison.** As mentioned in the previous section, we increase the number of parameters of the MoS model and name it MoS$^+$ to provide a fair comparison under the same number of parameters. To construct MoS$^+$, we increase the embedding size from 280 to 410. In Table 1, we notice that MoS$^+$ has a higher perplexity compared to MoS and CT-MoS, indicating that directly increasing model parameters cannot improve the performance in this case. Similar observation and results are also reported by (Yang et al., 2018). This ablation study shows that the improvements brought by CT-MoS are more than the mere grow of parameters.

---

[1] https://github.com/zihangdai/mos

Table 1: Performance Comparison on the Penn Treebank (PTB) dataset

| Model | #Param | Validation | Test |
|---|---|---|---|
| AWD-LSTM w/o finetune | 24M | 60.7 | 58.8 |
| AWD-LSTM | 24M | 60.0 | 57.3 |
| AWD-LSTM + dynamic evaluation | 24M | 51.6 | 51.1 |
| MoS w/o finetune | 22M | 58.08 | 55.97 |
| MoS | 22M | 56.54 | 54.44 |
| MoS + dynamic evaluation | 22M | 48.33 | 47.69 |
| $MoS^+$ w/o finetune | 24M | 59.72 | 57.43 |
| $MoS^+$ | 24M | 58.54 | 56.36 |
| $MoS^+$ + dynamic evaluation | 24M | 50.49 | 49.81 |
| CT-MoS w/o finetune | 24M | **56.95** | **54.69** |
| CT-MoS | 24M | **55.31** | **53.2** |
| CT-MoS + dynamic evaluation | 24M | **48.12** | **47.42** |

Table 2: Perfomrance Comparison on the WikiText-2 (WT2) dataset

| Model | #Param | Validation | Test |
|---|---|---|---|
| AWD-LSTM w/o finetune | 33M | 69.1 | 66.0 |
| AWD-LSTM | 33M | 68.6 | 65.8 |
| AWD-LSTM + dynamic evaluation | 33M | 46.4 | 44.3 |
| MoS w/o finetune | 35M | 66.01 | 63.33 |
| MoS | 35M | 63.88 | 61.45 |
| MoS + dynamic evaluation | 35M | **42.41** | **40.68** |
| CT-MoS w/o finetune | 45M | **65.25** | **62.21** |
| CT-MoS | 45M | **62.89** | **60.13** |
| CT-MoS + dynamic evaluation | 45M | 42.88 | 40.96 |

**Temperature Normalization Methods.** In addition to using the Softmax function in Equation 5 to normalize the temperature between $[\frac{\alpha}{\beta}, \frac{1+\alpha}{\beta}]$, we consider several alternative normalization methods: (a) $\lambda^{Tanh(\mu)}$, provided by (Lin et al., 2018), making the range of the temperature to be $(\frac{1}{\lambda}, \lambda)$, (b) $Tanh(\mu) + \lambda$, which generates a range of $(\lambda - 1, \lambda + 1)$, and (c) out method $(\sigma(\mu) + \alpha)/\beta$, where $\mu = f(x_{1..t-1}; \theta)^T \cdot W_{\tau_1} \cdot W_{\tau_2}$ is defined for simplicity. Results are shown in Table 3. For the conciseness, the three methods are evaluated on the PTB dataset without either fine-tuning or dynamic evaluation. The experimental results show that our proposed method outperforms other temperature normalization methods.

Table 3: Different methods for temperature normalization on the PTB dataset

| Model | hyper-parameter | range | #Param | Validation | Test |
|---|---|---|---|---|---|
| $\lambda^{Tanh(\mu)}$ (Lin et al., 2018) | $\lambda = 4$ | [1/4, 4] | 24M | 65.11 | 62.21 |
| $Tanh(\mu) + \lambda$ | $\lambda = 3$ | [2,4] | 24M | 61.35 | 58.89 |
| $(\sigma(\mu) + \alpha)/\beta$ | $\alpha = 1, \beta = 0.5$ | [2,4] | 24M | **56.95** | **54.69** |

**Effects of Loss Scaling.** As discussed in Section 3.3, the loss value may need to be scaled since applying the temperature may lead to smaller gradients. Here, we compare the results of whether applying loss scaling or not, the results shown in Table 4. Note that using loss scaling leads to better results, i.e. lower perplexity.

Table 4: Perplexity on PTB w/o finetune

| Model | #Param | Validation | Test |
|---|---|---|---|
| CT-MoS w/o loss scaling | 24M | 57.13 | 55.28 |
| CT-MoS | 24M | **56.95** | **54.69** |

## 4.5 ANALYSIS

**Temperature Change During Training.**   Going back to Figure 1(a) in the Introduction section, we have shown that contextual temperature is capable of generating a changing temperature as the training proceeds on a per vocabulary basis. This allows each vocabulary to have a flexible temperature schedule that is optimal to the model. Tokens that have higher temperatures are common words that don't convey much information. This suggests that the model is more confident about these words. Our method has the advantage of determining the schedule automatically, which is difficult to achieve in the traditionally fixed temperature scheduling method.

**Average Temperature Over Word Positions.**   We also want to dive deeper into Figure 1(b), which shows the mean and confidence interval of the temperature over different positions. We group sentences whose length ranges from 15 to 25. For sentences whose length are greater than 15, we pick first 5 tokens, middle 5 tokens, and last 5 tokens to form a new sentence. At the beginning of a sentence, temperatures are usually high to smooth probabilities of the Softmax. This is the region where the model has little confidence since there is too little information in the history. As the history builds up over time, the model becomes more confident and the temperature begins to cool down, making the probability distribution spiky and the model more confident. The confidence intervals of the temperature also become tighter as the length of the history increases.

**Word Frequencies And Temperature Change.**   Evidence from Figure 1(a) suggests that only a small fraction of tokens have significantly larger temperatures. However, even small changes might deliver large impacts to the performance due to the facts that these tokens might be used more often than others. To further analyze, we present an analysis on the weighted temperature in Figure 3(a), that is, the temperature is multiplied by its frequency. Here we see that for the majority of words, even small changes in temperatures might have great effects as many of them occur fairly frequently.

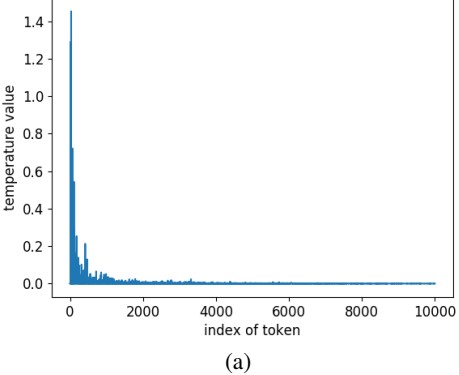

(a)

Figure 3: (a) The weighted temperature over the token index ranked by frequency. Here the weighted temperature refers to the absolute temperature multiplied by the frequency of the word in the corpus.

**Case Studies.**   We conduct several case studies on PTB to visualize the effect of contextual temperature. Table 5 shows the comparison of CT-MoS v.s. MoS. We highlight two differences between CT-MoS and MoS, annotated using red and blue colors. In the first spot, we see that both CT-MoS and MoS fail to predict the correct answer "single-a-1", which refers to a rating for securities. The

CT-MoS model predicts "triple-a", which is not the same as the reference but is much closer to the answer, since "triple-a" refers to the highest rating for securities. MoS, on the other hand, predicts <unk>, which deviates too much from the ground truth. Taking a close look at the temperature, the word "triple-a" has a temperature of $2 + 8.34 \times 10^{-5}$, which is a bit smaller than that of the <unk>, whose temperature is $2 + 8.6 \times 10^{-5}$. This contributes to the factor that the model chooses "triple-a" over <unk>. Another example is illustrated by the prediction of the word "standard". Here the temperature of "standard" is smaller than that of "s&p", making the model more likely to predict the prior word. We refer readers to Appendix B for more samples.

Table 5: Analysis of Model Performance on a Sample from the PTB Dataset

| Reference | rated **single-a-1** by moody 's investors service inc. and single-a by **standard** & poor 's corp. ... | | | |
|---|---|---|---|---|
| CT-MoS | rated **triple-a** by moody 's and service inc. and <unk> by **standard** & poor 's corp. ... | | | |
| MoS | rated <**unk**> by moody 's and service inc. and <unk> by **s&p** & poor 's corp. ... | | | |
| **CT-MoS top-4** | **triple-a 0.34** | single-a-2 0.2 | single-a-1 0.15 | single-a-3 0.11 |
| **MoS top-4** | <**unk**> **0.28** | **triple-a 0.27** | single-a-2 0.24 | single-a-1 0.1 |
| $\tau(1e^{-5} + 2)$ | **triple-a 8.34** | | <**unk**> **8.60** | |
| **CT-MoS top-4** | **standard 0.53** | **s&p 0.22** | moody 0.17 | dow 0.02 |
| **MoS top-4** | **s&p 0.4** | moody 0.23 | **standard 0.19** | <unk> 0.03 |
| $\tau(1e^{-5} + 2)$ | **standard 11.2** | | **s&p 11.4** | |

Another indicator of how contextual temperature works is to look at the change of the temperature across different positions in a sentence. In Table 6, we visualize the occurrence of the word "mortage" and its temperature. Here we see that as the position changes, contextual temperature chooses a different value for each of the position, adjusting its confidence of model's belief. As we analyze before, a general trend is that words appearing early in a sentence get larger temperatures while those approaching the end of the sentence get smaller temperatures.

Table 6: Analysis of Temperature for Same Words at Different Positions

| CT-MoS | loan **mortgage(1)** corp freddie mac posted posted yields on 30-year **mortgage(2)** commitments for delivery within N days <eos> N N standard conventional fixed-rate **mortgages(3)** N N N rate rate capped one-year adjustable rate **mortgages(4)** <eos> source telerate systems inc <eos> federal national **mortgage(5)** association fannie mae posted posted yields on N year **mortgage(6)** commitments for delivery within N days priced at par N N N standard conventional fixed-rate **mortgages(7)** N ... |
|---|---|
| $\tau(1e^{-5} + 2)$ | (1) 18.9   (2) 19.3   (3) 20.1   (4) 19.2   (5) 18.9   (6) 19.2   (7) 18.2 |

## 5 CONCLUSION AND FUTURE WORK

In this paper, we have proposed contextual temperature, a generalized and effective approach able to assigning individual optimal temperature of each class, by changing temperature based on the history of the context. Contextual temperature is parameterized using a deep neural network, and generates a unique schedule for each vocabulary to compute the corresponding optimal temperature. Experiments on the language modeling datasets achieve significantly better performances. In the future, our work opens up potential new research directions along the line of fully automated temperature mechanism to explore the implementation of contextual temperature in various NLP tasks such as summarization, machine translation, and dialogue generation.

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

## A    EXPERIMENT RESULTS OF CT-MOS, MOS, TRANSFORMER-XL AND GPT-2

Below shows the experiment results of CT-MoS, MoS, Transformer-XL and GPT-2 on the Penn Treebank dataset. Lower perplexity represents better performance. Please note that the GPT model works on a very different setting than the ones found in the mainstream language model research. In the GPT paper, it utilizes a large dataset that is collected outside the domain of the language modeling.

Table 7: Performance Comparison on the PTB dataset

| Model | #Param | Validation | Test |
|---|---|---|---|
| CT-MoS w/o finetune | 24M | 56.95 | 54.69 |
| CT-MoS | 24M | 55.31 | 53.2 |
| CT-MoS + dynamic evaluation | 24M | 48.12 | 47.42 |
| MoS w/o finetune | 22M | 58.08 | 55.97 |
| MoS | 22M | 56.54 | 54.44 |
| MoS + dynamic evaluation | 22M | 48.33 | 47.69 |
| Tansformer-XL | 24M | 56.72 | 54.52 |
| GPT-2 | 117M | - | 65.85 |
| GPT-2 | 345M | - | 47.33 |
| GPT-2 | 762M | - | 40.31 |
| GPT-2 | 1542M | - | 35.76 |

## B    MORE SAMPLES FROM PTB

Table 8: More samples from PTB.

| Reference | these rate indications are n't directly comparable lending practices vary widely by location <eos> treasury bills <eos> **results** of the tuesday ... | | | |
|---|---|---|---|---|
| CT-MoS | these rate indications are n't directly comparable lending practices vary widely by location <eos> treasury bills results **results** of the monday ... | | | |
| MoS | these rate indications are n't directly comparable lending practices vary widely by location <eos> treasury bills results **treasury** of the monday ... | | | |
| **CT-MoS top-4** | **results 0.81** | **treasury 0.07** | a 0.01 | bonds 0.01 |
| **MoS top-4** | **treasury 0.64** | **results 0.09** | the 0.04 | N 0.02 |
| $\tau(1e^{-6}+2)$ | **results 8.11** | | **treasury 8.58** | |

| Reference | corporate loans at large u.s. money center commercial banks <eos> federal funds <eos> N N **N** ... | | | |
|---|---|---|---|---|
| CT-MoS | corporate loans at large u.s. money center commercial banks <eos> federal funds N N N **N** ... | | | |
| MoS | corporate loans at large u.s. money center commercial banks< eos> federal funds N N N **high** ... | | | |
| **CT-MoS top-4** | **N 0.45** | **high 0.41** | <eos> 0.05 | low 0.04 |
| **MoS top-4** | **high 0.46** | **N 0.41** | <eos> 0.06 | and 0.02 |
| $\tau(1e^{-4}+2)$ | **N 1.56** | | **high 2.01** | |

| Reference | a share compared with a net loss of $ N million last year after a **loss** from discontinued operations ... | | | |
|---|---|---|---|---|
| CT-MoS | a share <eos> with $ loss loss of $ N million or year <eos> the **loss** of the operations ... | | | |
| MoS | a share <eos> with $ $ loss of $ N million or year <eos> the **$** of the operations ... | | | |
| **CT-MoS top-4** | **loss 0.24** | **$ 0.20** | N 0.06 | <unk> 0.06 |
| **MoS top-4** | **$ 0.11** | **loss 0.09** | one-time 0.07 | N 0.07 |
| $\tau(1e^{-4}+2)$ | **loss 1.40** | | **$ 1.62** | |

| Reference | in the **nine** months <unk> 's net rose N N to $ N million ... | | | |
|---|---|---|---|---|
| CT-MoS | the the **nine** months the said net income N N to $ N million ... | | | |
| MoS | the the **first** months the said net income N N to $ N million ... | | | |
| **CT-MoS top-4** | **nine 0.17** | third 0.10 | year-ago 0.09 | year-earlier 0.09 |
| **MoS top-4** | **first 0.14** | **nine 0.12** | third 0.12 | year-ago 0.09 |
| $\tau(1e^{-5}+2)$ | **nine 7.37** | | **first 7.94** | |

