# OpenReview forum: "Contextual Temperature for Language Modeling"
_ICLR.cc/2020/Conference — Reject_

### Official Review · AnonReviewer1 · 2019-10-22
**Official Blind Review #1**

**Rating:** 6

**Review:**

This work proposes a learned and context dependent way to calculate the temperatures for the softmaxes. More specifically, a low-rank affine-transformation, taking the hidden state at the current step as input, is used to calculate scalar weighting for every token in the vocabulary. The method is very general, and can be used in combination with other techniques in tasks such as language modeling and text generation. Experiments on language modeling with Penn TreeBank and WikiText-2 show that the proposed method yields strong performance.

Overall I found the paper well-motivated and easy to follow. The empirical results are solid and strong. The analysis is also interesting. I vote for an acceptance, if the authors can polish the writing.

Details:

- Eq. 5. The temperature scalar for each token competes with each other, since they are calculated with a softmax (and then rescaled). Another way is to use, e.g., a sigmoid function. Can the authors explain the motivation behind the use of softmax?

- Another view of the proposed method is that it learns a context-dependent weighting of the tokens in the vocabulary, such that "important" tokens (those with smaller \tau) receive more gradient updates. Can the authors comment on this? Also, I don't see the thermodynamics connection and find calling the proposed method `temperature` a bit misleading.

- Adding onto above. [1] discusses the low-rank bottleneck of using a single softmax. Since elementwise matrix product can blow up the rank, how do the authors think the proposed method can serve as a more efficient way to deal with the softmax bottleneck?

- Last but not least, the paper can be improved a lot if the authors can thoroughly polish the writing.


[1] Breaking the Softmax Bottleneck: A High-Rank RNN Language Model. https://arxiv.org/abs/1711.03953

**Experience Assessment:**

I have published in this field for several years.

**Review Assessment: Checking Correctness Of Derivations And Theory:**

N/A

**Review Assessment: Checking Correctness Of Experiments:**

I assessed the sensibility of the experiments.

**Review Assessment: Thoroughness In Paper Reading:**

I read the paper at least twice and used my best judgement in assessing the paper.

---

> ### Author Response · Authors · 2019-11-13
> **Response to AnonReviewer1 [1/2]**
>
> First of all, we thank the reviewer for the constructive feedback.
>
> (Q1) Eq. 5. The temperature scalar for each token competes with each other, since they are calculated with a softmax (and then rescaled). Another way is to use, e.g., a sigmoid function. Can the authors explain the motivation behind the use of softmax?
>
> (A1) We agree with the reviewer that using a softmax to control the temperature of each token will make these temperatures compete with each other (since they have to sum up to 1). As the reviewer suggested, we conduct more experiments to compare softmax with tanh and sigmoid. Experiment results show that using softmax achieves the lower perplexity compared to the other two functions—softmax: 54.69, sigmoid: 57.74, and tanh: 58.89 (on the test set of the PTB dataset). We conjecture that the relationship among different tokens represents a certain kind of competitiveness as only a few tokens share similar semantics as the ground-truth token should be generated in a sentence. We appreciate the suggestion from the reviewer.
>
> (Q2) Another view of the proposed method is that it learns a context-dependent weighting of the tokens in the vocabulary, such that "important" tokens (those with smaller \tau) receive more gradient updates. Can the authors comment on this?
>
> (A2) We appreciate the reviewer's insights on the relationship between the importance of tokens and the magnitude of the gradients from the contextual temperature. As each token is represented by an embedding that has multiple dimensions, we believe ‘more gradient updates’ mentioned by the reviewer actually refers to larger gradient norms. If that's the case, we generally agree with the reviewer's insights.
>
> In the paper, we observe that common tokens ('<eos>’, ‘of’, ‘the’, ...) receive the temperature that increases dramatically during the course of training (please refer to Figure 1a in the paper), which effectively scales down the corresponding logits. We refer these tokens as ‘unimportant’ tokens to contrast the rest of tokens (referred to as ‘important’ tokens). We then calculate the average gradient norm of ‘important tokens’ and repeat the same procedure for ‘unimportant’ tokens. The results are provided below.
>
> To confirm the reviewer's conjecture, we calculated the average norm of gradients with respect to the embedding parameters. We calculated the results separately for the case when `important tokens' are the ground truth and the case when `unimportant tokens` are ground truth. Results are shown in the next table.
>
> We note that, when the ground truth belongs to ‘important tokens’, the average gradient norm of ‘important tokens’ is larger. Same story for unimportant tokens: When the ground truth belongs to unimportant tokens, the average gradient norm of unimportant tokens is larger. In other words, depending on the ground truth belonging to important or unimportant tokens, applying contextual temperature seems to make the corresponding type of tokens receive a larger gradient norm.
>
> (1) If the ground truth belongs to “important tokens”, then
> gradient norm of "important" tokens	  	avg. \tau of “important” tokens
> ———————————————————————————————————————
>              0.260164  (a)			                                            2.0001514
>
> gradient norm of "unimportant" tokens	 avg. \tau of “unimportant” tokens
> ———————————————————————————————————————
>              0.045674  (b)                                                                2.0696018
>
> (2) If the ground truth belongs to “unimportant tokens”, then
> gradient norm of "important" tokens		avg. \tau of “important” tokens
> ———————————————————————————————————————
>              0.280090  (c)			                                            2.0001671
>
> gradient norm of "unimportant" tokens	 avg. \tau of “unimportant” tokens
> ———————————————————————————————————————
>              0.491596  (d)                                                                2.047023
>
> Here, each of the entries is calculated as the following, where y represents samples, L2(*) represents the norm function, ∇(y,Θ) represents the gradient vector of sample y and parameter Θ, Θ represents a parameter and E is the expectation function.
>
> a = E_{y in important} E_{Θ in important} L2(∇(y,Θ))
> b = E_{y in important} E_{Θ in unimportant} L2(∇(y,Θ))
> c = E_{y in unimportant} E_{Θ in important} L2(∇(y,Θ))
> d = E_{y in unimportant} E_{Θ in unimportant} L2(∇(y,Θ))

---

> ### Author Response · Authors · 2019-11-13
> **Response to AnonReviewer1 [2/2]**
>
> (Q3) Also, I don't see the thermodynamics connection and find calling the proposed method `temperature` a bit misleading.
>
> (A3) We follow the naming convention in related previous works we’ve known [1, 2, 3] and call our method “temperature”. As mentioned in [2], the connection between temperature scaling (in deep learning domain) and thermodynamics can be found in statistical mechanics [4]. We are willing to hear if there are any advice about the naming.
>
> [1] Geoffrey Hinton, Oriol Vinyals, and Jeff Dean. Distilling the knowledge in a neural network. 2015.
> [2] C. Guo, G. Pleiss, Y. Sun, and K. Q. Weinberger. On calibration of modern neural networks. 2017.
> [3] Zhiting Hu, Zichao Yang, Xiaodan Liang, Ruslan Salakhutdinov, and Eric P. Xing. Toward controlled generation of text. 2018.
> [4] Jaynes, Edwin T. Information theory and statistical mechanics. 1957.
>
> (Q4) Adding onto above. [1] discusses the low-rank bottleneck of using a single softmax. Since elementwise matrix product can blow up the rank, how do the authors think the proposed method can serve as a more efficient way to deal with the softmax bottleneck?
>
> (A4) We thank the reviewer for the great feedback. According to rank inequality( R(A०B)≤R(A)R(B) ), element-wise matrix product indeed will potentially increase the rank. This is a new theoretical direction for the proposed contextual temperature, and we will study more in-depth in this direction and update the manuscript when having concrete conclusion and/or findings. Due to the limited time of ICLR rebuttal, we are not able to finish the analysis before the deadline. However, we will keep working on finding the evidence of this conjecture as the reviewer suggested.
>
> (Q5) Last but not least, the paper can be improved a lot if the authors can thoroughly polish the writing.
>
> (A5) Thank you for the advice. We have identified several spots in the paper that we can further polishing. Additionally, we have also improved the introduction section as well as the analysis section. We will keep looking for potential issues in writing. Here we list a few of changes we have made:
> In Abstract:  co-adopt => co-adapt
> In Introduction:  exiting work => existing methods
> "explored the vocabulary differences when adjusting temperature"  => "explored the differences among vocabulary tokens when adjusting temperature"
> "tends to be heating up" => "tends to heat up"
> "This suggests that temperature mechanism helps to promote stochasticity early in
> the sentence while suppressing uncertainties when the context gets longer" => "This suggests that the temperature mechanism helps promote stochasticity early in the sentence, and suppress uncertainties when the context gets longer."
> dealing with these phenomenons => dealing with these phenomena.

---

### Official Review · AnonReviewer2 · 2019-10-23
**Official Blind Review #2**

**Rating:** 3

**Review:**

This paper proposed a contextual temperature scaling to improve language modeling. The temperature model is parameterized using a deep neural network. Experiments on the language modeling datasets show some effects of the method.

The idea of dynamic temperature scaling has been tried in other works and tasks (e.g., attended temperature scaling). The paper parameterizes this mechanism with DNNs for the language model.  Though the idea looks interesting, it fails to explain why the scaling is better than other dynamic temperature scaling frameworks.

The experiments are not solid. The baseline only includes Mos, which is not very strong. To validate whether this approach works with other LM of high-order attention or self-attention, a better baseline model is required (e.g., transformer, GPT). I would like to see this technique can help either NLU or NLG tasks, instead of just pure modeling. The case analysis section needs more examples instead of just cherry-picking few.

**Experience Assessment:**

I have read many papers in this area.

**Review Assessment: Checking Correctness Of Derivations And Theory:**

N/A

**Review Assessment: Checking Correctness Of Experiments:**

I carefully checked the experiments.

**Review Assessment: Thoroughness In Paper Reading:**

I read the paper at least twice and used my best judgement in assessing the paper.

---

> ### Author Response · Authors · 2019-11-13
> **Response to AnonReviewer2**
>
> First of all, we thank the reviewer for the feedback.
>
> (Q1) The idea of dynamic temperature scaling has been tried in other works and tasks (e.g., attended temperature scaling). The paper parameterizes this mechanism with DNNs for the language model.  Though the idea looks interesting, it fails to explain why the scaling is better than other dynamic temperature scaling frameworks.
>
> (A1) We appreciate the feedback from the reviewer. To the best of our knowledge, we are the first work to learn a different temperature for each token based on the context. We have done a comprehensive survey on related works, but we haven’t find any similar work. To answer the reviewer's question, the proposed method distinguishes from the other attended temperature scaling since attended temperature scaling paper learns a temperature that is universal for all the classes (tokens). Contextual temperature, on the other hand, learns a different temperature for each token and is thus a more general approach. To make it more clear to readers on the differences between our method and other related ones, we plan to modify the paper to emphasize the differences.
>
> (Q2) The experiments are not solid. The baseline only includes Mos, which is not very strong. To validate whether this approach works with other LM of high-order attention or self-attention, a better baseline model is required (e.g., transformer, GPT).
>
> (A2) We would like to point out that MoS is the state-of-the-art model on language modeling on the Penn Treebank dataset and WikiText-2 dataset. The Transformer-XL model, which is based on the transformer architecture, actually performs worse than the MoS model in these two datasets. To make the comparison clear, below we've put a summary of the comparisons between these baselines and our approach. Although the Transformer-XL model performs worse than MoS in the paper, we do agree that it should be added as a comparison to the paper. We will revise it in an updated version.
>
> Model | validation ppl | test ppl
> CT-MoS (ours)                              | 55.31 | 53.20
> MoS                            			| 56.54 | 54.44
> Transformer-XL                            | 56.72 | 54.52
> GPT-2 (w/ extra training data and significantly larger model params) |-| 35.76
>
> Finally, the GPT model works on a very different setting than the ones found in the mainstream language model research. In the GPT paper, it utilizes a large dataset that is collected outside the domain of the language modeling. We argue that a comparison between GPT and the other baselines would be unfair as the standard setting of language modeling do not use additional datasets. As the proposed contextual temperature method aims at improving language model in the standard setting without the use of additional dataset, we believe that it would be more appropriate to compare against baselines under the same setting.
>
> (Q3) I would like to see this technique can help either NLU or NLG tasks, instead of just pure modeling.
>
> (A3) We would like to point out that the goal of our paper is to study language modeling other than its performance on the downstreaming NLU or NLG tasks. It is true that many language models such as BERT and XLNet are designed specifically for boosting the performance of downstream NLU and NLG tasks, models that study language modeling such as MoS focus purely on the performance of the language model itself. To this end, we feel it is important to separate these two groups of research as they fundamentally serve as different goals. We will clarify the differences on an updated version of the paper.
>
> (Q4) The case analysis section needs more examples instead of just cherry-picking few.
>
> (A4) We have provided more examples in Appendix A. Hopefully that can provide more insights on the methods. We will opensource the codebase upon the acceptance of this paper to allow the examinations of more examples.

---

### Official Review · AnonReviewer3 · 2019-11-06
**Official Blind Review #3**

**Rating:** 3

**Review:**

This paper presents a strategy to automatically adjust the temperature scaling based on the context of words in a sentence for NLP. Experiments demonstrate that this approach can significantly improve perplexity scores on several datasets popular for NLP.

NLP is not an area of research I'm very familiar with so this review is limited to my understanding of temperature scaling as a general technique to improve learning. As described in the paper, temperature scaling is a type of hyper-parameter estimation that adjusts the sensitivity of the softmax function as training evolves. The paper proposes to learn a function that given context, adjust the temperature automatically. This can be seen as a meta-learning method.

I believe this can be a useful technique but before considering such an approach as a general strategy, more theoretical insights should be provided. The authors report on ablation studies that demonstrate some empirical benefits. However, until I see more theoretical analysis on how the method improves convergence or lead to better losses by smoothing out the output of the objective function, I remain skeptical of the usefulness of this as a general training method.

**Experience Assessment:**

I do not know much about this area.

**Review Assessment: Checking Correctness Of Derivations And Theory:**

I assessed the sensibility of the derivations and theory.

**Review Assessment: Checking Correctness Of Experiments:**

I did not assess the experiments.

**Review Assessment: Thoroughness In Paper Reading:**

I made a quick assessment of this paper.

---

> ### Author Response · Authors · 2019-11-13
> **Response to AnonReviewer3**
>
> First of all, we thank the reviewer for the feedback.
>
> Temperature scaling as a technique to control the smoothness of the softmax output is widely used in NLP, as can be seen from the large body of literature that we have surveyed in our paper. We have examined each of them but we found it generally difficult to justify the method from a theoretical perspective: either on its convergence or the property of the loss function that it leads to. However, even if theoretical analysis is difficult, its empirical performance of temperature has been verified by a large body of work in NLP.
>
> Similar to the situation of the temperature in general, our model, which builds on a highly nonlinear transformation of inputs, is difficult to generate theoretical guarantees. However, given the large amount of empirical evidence, it is unlikely that the effectiveness of this approach is a coincidence. We believe the significance of the proposed contextual temperature is that it provides a more general view of the temperature mechanism. And its effectiveness can be demonstrated in a wide range of NLP tasks. If the reviewer has any further suggestions on the theoretical analysis, we would love to know.

---

### Author Response · Authors · 2019-11-15
**Paper Update**

We appreciate the constructive feedback of every reviewer. We have thoroughly refined the paper: grammar errors are corrected, sections including abstract, introduction and experiments are retouched, and appendix is added to provide more clear explanations.

---

### Decision · Program_Chairs · 2019-12-19

**Decision:**

Reject

**Comment:**

With an average post author response score of 4 - two weak rejects and one weak accept, it is just not possible for the AC to recommend acceptance. The author response was not able to shift the scores and general opinions of the reviewers and the reviewers have outlined their reasoning why their final scores remain unchanged during the discussion period.